# Multi-UAV Data Collection and Path Planning Method for Large-Scale Terminal Access

**DOI:** 10.3390/s23208601

**Published:** 2023-10-20

**Authors:** Linfeng Zhang, Chuhong He, Yifeng Peng, Zhan Liu, Xiaorong Zhu

**Affiliations:** 1Institute of Mobile and Terminal Technology, China Telecom Research Institute, Guangzhou 510630, China; zhanglinf@chinatelecom.cn (L.Z.); liuz22@chinatelecom.cn (Z.L.); 2School of Communication and Information Engineering, Nanjing University of Posts and Telecommunications, Nanjing 210003, China; 1022010105@njupt.edu.cn (C.H.); 16670602602@163.com (Y.P.)

**Keywords:** UAV, data collection, equalize data volume, route planning, time minimization

## Abstract

In the context of the relentless evolution of network and communication technologies, the need for enhanced communication content and quality continues to escalate. Addressing the demands of data collection from the abundance of terminals within Internet of Things (IoT) scenarios, this paper presents an advanced approach to multi-Unmanned Aerial Vehicle (UAV) data collection and path planning tailored for extensive terminal accessibility. This paper focuses on optimizing the complex interplay between task completion time and task volume equilibrium. To this end, a novel strategy is devised that integrates sensor area partitioning and flight trajectory planning for multiple UAVs, forming an optimization framework geared towards minimizing task completion duration. The core idea of this work involves designing an innovative k-means algorithm capable of balancing data quantities within each cluster, thereby achieving balanced sensor node partitioning based on data volume. Then, the UAV flight trajectory paths are discretely modeled, and a grouped, improved genetic algorithm is used to solve the Multiple Traveling Salesman Problem (MTSP). The algorithm introduces a 2-opt optimization operator to improve the computational efficiency of the genetic algorithm. Empirical validation through comprehensive simulations clearly underscores the efficacy of the proposed approach. In particular, the method demonstrates a remarkable capacity to rectify the historical issue of diverse task volumes among multiple UAVs, all the while significantly reducing task completion times. Moreover, its convergence rate substantially outperforms that of the conventional genetic algorithm, attesting to its computational efficiency. This paper contributes an innovative and efficient paradigm to improve the problem of data collection from IoT terminals through the use of multiple UAVs. As a result, it not only augments the efficiency and balance of task distribution but also showcases the potential of tailored algorithm solutions for realizing optimal outcomes in complex engineering scenarios.

## 1. Introduction

In Wireless Sensor Networks (WSNs), a large number of sensor nodes are deployed to detect parameters in the environment, including temperature, humidity, smoke concentration, etc. [1]. These parameters exist in the form of direct bit-streams, voice, and video. WSNs have enabled new applications, including smart electricity, water supply networks, and intelligent transportation [2]. Sensors need to transmit the data collected in the network to the monitoring system. However, due to the limited power and small signal coverage of these sensors, the usual method is to deploy a full-coverage network to receive these data, which greatly increases the cost of network construction and operation [3]. In the absence of these infrastructures, how to upload and process data in a timely manner, as well as how to effectively collect these data, has become a problem. Furthermore, the presence of a large number of sensor nodes poses challenges for collecting massive amounts of data. IoT data collection demands low energy consumption, low latency, and high reliability, and UAV technology offers new opportunities to meet these requirements. UAVs can be dynamically deployed at low cost, making them suitable as communication hubs between WSNs and the monitoring system [4]. Due to their wide coverage, high rate, and strong mobility, UAVs have gradually attracted everyone’s attention. UAVs inherently possess exceptional maneuverability, making them an ideal tool for addressing data transmission needs. Leveraging UAVs not only reduces energy consumption in IoT data transmission but also enables wireless charging for sensors, extending network lifespans. Additionally, UAVs can collect data when in proximity to sensors, further reducing data transmission energy consumption. The combination of UAVs and IoT, especially in complex, adverse, or remote environments, facilitates efficient and timely data collection. The emergence of multi-UAV collaborative systems further enhances task efficiency, expands operational domains, and increases scalability. These systems can tackle increasingly complex task demands across various fields such as search and rescue, surveillance, disaster monitoring, transportation, public safety, and defense, offering broad application prospects. Consequently, UAV technology holds significant potential for meeting future needs for collecting vast amounts of data in the IoT.

In WSNs, in order to reduce the cost and prolong the network lifetime, the UAV can be used as the air mobile base station/relay for data collection. The authors in [5] suggested using UAVs to achieve efficient data collection from IoT devices. Specifically, considering the differences between IoT devices in terms of data generation rate, the data collection task is modeled as a UAV routing problem with different time windows and service time requirements. In [6], the authors divided WSNs into regular hexagonal network units and applied UAVs for sensor localization and data collection. This paper proposes a sensor localization scheme based on Received Signal Strength (RSS). In [7], the authors explored optimizing data collection in WSNs using multiple UAVs. The research focused on minimizing the total task time for UAVs, which can communicate with sensors while flying or hovering, and proposed a jointly optimized UAV-sensor association mechanism and data collection method for UAVs. The authors in [8] focused on the scenario of a UAV-enabled wireless sensor network with delay-tolerant and delay-sensitive applications. This paper aimed to optimize UAV trajectory design and ground nodes’ transmit power allocation to maximize average data-rate throughput and minimize transmission outage probability. In [9], the authors presented an innovative framework for efficient data collection from WSNs using a combination of UAVs and Unmanned Ground Vehicles (UGVs). The UGVs were equipped with backup batteries and moved along with the UAVs. The optimization objective was to minimize the task time for a full round of data collection, which was solved by a heuristic path planning algorithm. The authors in [10] explore multicast communication in a satellite and UAV integrated network using Rate-Splitting Multiple Access (RSMA), with a focus on supporting massive access of IoT devices (IoTDs) in a content delivery scenario, aiming to achieve interference suppression, spectral efficiency, and hardware complexity. The authors in [11] model the UAV communications with jittering effects, analyze the effect of the UAV movement and attitude variation on the channel, and propose an efficient channel estimation method. Nevertheless, they did not consider the specific application of UAVs in IoT scenarios, nor did they consider the collaboration between multiple UAVs. In [12], the authors developed a many-objective optimization model for deploying multiple UAV onboard cameras within a 3D terrain environment represented by triangular mesh data. This model accounts for various objective functions, including coverage rate, point-level clarity, uniform clarity, and resource utilization rate. Their primary focus was on addressing the static deployment of UAVs without delving into the trajectory planning aspects of UAVs. The authors in [13] utilize the UAV as the relay and edge computing node to support the decision-making system in intelligent transportation networks, especially when faced with traffic congestion or when RoadSide Units (RSUs) are beyond the communication range. Additionally, their paper introduces a deep reinforcement learning-based channel allocation and task offloading strategy to enhance data transmission efficiency. However, it is worth noting that their paper mainly considers the resource allocation and task offloading decision-making problem of a single UAV used in the vehicle network without considering the collaboration between multiple UAVs and the dynamic planning of flight trajectories. Moreover, several previous studies concentrated on optimizing the trajectories, postures, and velocities of autonomous vehicles, or UAVs, primarily from the perspective of automation control rather than communication networks [14,15,16]. Additionally, certain other earlier works delve into a range of associated topics, encompassing cooperative UAV positioning applications [17], RFID-assisted human motion recognition [18], spatio-temporal traffic pattern learning and prediction for private car weekend gatherings [19], vehicle-to-infrastructure (V2I) time delay estimation at junctions [20], aircraft flight trajectory prediction based on wavelet transform [21], and maneuvering decision-making for multi-Underwater Unmanned Vehicles (UUVs) in complex underwater environments [22]. Nevertheless, these studies did not incorporate considerations of balanced node clustering and flight trajectory planning for data collection tasks within the context of multi-UAV collaboration in WSNs.

Due to the large number of terminals in WSNs and the fact that many sensors do not have storage capabilities, it is unrealistic for UAVs to communicate directly with all sensor nodes. Therefore, a clustering method can be adopted to cluster all sensor nodes, and a cluster head node with a storage function is deployed in the center of each cluster. The remaining sensors first pass the collected data to the cluster head node for storage and wait for the UAV to fly close to the cluster head node to pass this information to the UAV, and then the UAV returns to the monitoring center. Although UAVs have many advantages, they also have the disadvantage of limited energy consumption, which requires charging after working for a period of time.

The study to address the multi-objective global path planning challenge for UAV-assisted sensor data collection in the context of expanding IoT-generated business data was conducted in [23]. It leveraged deep reinforcement learning to decompose the problem into subproblems represented by neural networks. An actor-critic algorithm and modified pointer network were employed to solve each subproblem, resulting in a Pareto front for the path planning solution, effectively maximizing data collection and minimizing UAV flight time.

In [24], the authors addressed the challenge of optimizing UAV path planning for data collection from ground-fixed devices deployed to monitor forest pests and wildlife. It introduced two methods: a chaotic initialization and co-evolutionary algorithm for two-point path planning and a UAV path planning approach based on simulated annealing for multi-point path planning.

The path planning problem for UAV data collection was addressed by the authors in [25]. They did it by dividing it into global and local planning stages. Global planning was formulated as an orientation problem, merging the knapsack and traveling salesman problems, which were then solved using deep learning’s pointer network. For local planning, the UAV’s flight path was determined by a deep Q network using the RSS from sensor nodes.

In [26], the authors presented an innovative asynchronous UAV path planning mechanism for multi-objective UAV operation in large-scale WSNs. The proposed approach employed a multipurpose fitness function and Particle Swarm Optimization (PSO) algorithm. By taking into account location-dependent communication link quality, sensor density, and the next UAV position, along with considering several UAV operational constraints, the optimal UAV positions and the movement schedule for each UAV are determined.

Furthermore, in the realm of optimization, previous studies [27,28,29] tackled complex non-convex optimization problems within contexts associated with multibeam satellite systems. They employed diverse methodologies, including the Charnes-Cooper approach coupled with iterative search algorithms, the sequential convex approximation method, singular value decomposition, uplink-downlink duality, Taylor expansion, penalty function techniques, and successive convex approximation methods, among others, to effectively address these intricate challenges.

This paper addresses the multi-UAV uplink data collection scenario within a large-scale IoT environment, focusing on the reception of data transmitted by IoT terminals. Considering the balanced clustering of IoT terminals and the trajectory planning of multiple UAVs with interference management, an optimization problem minimizing the task completion time was constructed. Firstly, we apply the traditional distance-based k-means algorithm to cluster IoT terminals, and based on this, we have designed an algorithm to adjust cluster division results and cluster head node positions in order to balance the task data volume and data types within each cluster. Secondly, the trajectory design problem of multi-UAV data collection was solved, and the flight path of the UAV was discretized, modeling the problem as a MTSP. The deformation of this MTSP was solved by a group-improved genetic algorithm (GA). Finally, the simulation experiments of our scenarios and methods were proposed, and the experimental results were analyzed. The novelty of this study lies in proposing an innovative and efficient solution for optimizing the completion time of data collection tasks in the context of large-scale IoT terminal access scenarios. This solution involves using an improved k-means algorithm to cluster sensor nodes based on data volume balance. Additionally, it utilizes a grouping improved genetic algorithm based on the 2-opt optimization operator to plan the data collection flight paths for multiple UAVs. With the algorithm presented in this paper, even in the absence of infrastructure such as base stations, the high maneuverability of UAVs and the collaborative capabilities among multiple UAVs can be leveraged to meet the data transmission needs of a massive number of terminal devices in sensor networks. This is of significant importance for the efficient collection of data from large-scale IoT terminal devices.

## 2. System Model

In the data collection task cycle, the UAV flight trajectory is planned to collect data from all sensors. After the task is completed, the UAV needs to return to the starting point for charging for the next round of data collection. Our goal is to minimize the time of data collection in this cycle as well as to balance the working time of each UAV, and the network scenario diagram is shown in Figure 1. Suppose there are N sensors randomly distributed within the network, and the types of data collected by these sensors include temperature, humidity, smoke concentration, noise level, and video recording, among others. The forms of data collected include text, voice, video signals and so on. These sensors have low power and lack infrastructure such as gateways in the network, so U UAVs need to be dispatched as aerial mobile collection points to collect data from these sensors. Most sensors have no storage capacity, so K cluster head nodes, K≫U, can be deployed in the network. It is assumed that the cluster head node has a transmit power of $P_{c}$ and the other nodes have a transmit power of $P_{s}$, with $P_{c}\gg P_{s}$. All the sensor nodes are divided into K clusters, and each sensor must be bound to a cluster head node $c_{i}$. The data are first passed to the cluster head node for storage and collection by the UAV. The received Signal-to-Noise Ratio (SNR) of cluster head node $c_{i}$ can be expressed as follows:(1)$$\gamma_{c_{i}}=\frac{P_{s}d_{ji}^{-\alpha }}{\sigma^{2}},\forall i=1,\ldots ,K,\forall j=1,\ldots ,N$$where $d_{ji}$ is the distance between sensor $s_{j}$ and cluster head node $c_{i}$, α is the path loss exponent, and $\sigma^{2}$ is the noise power. For $c_{i}$, its received SNR needs to meet the threshold $\gamma_{th}^{C}$ in order to accurately receive the data transmitted by other sensors. When the UAV flies over the close cluster head node $c_{i}$, it communicates with $c_{i}$, so as to collect data from sensors in the i-th cluster.

Further more, Tu is used to denote the completion time for the UAV u to complete the data collection task, where u=1,2,…,U. To simplify the presentation, suppose that all UAVs are fixed flying at altitude H, and the horizontal trajectory position of the UAV u can be represented as qu(t)=xu(t),yu(t)T∈ℝ2×1. Suppose that Si is the set of all sensors in the i-th cluster, and W=wi=[xi,yi]T∈ℝ2×1∀i represents the set of the positions of all cluster head nodes.

The communication between the UAV and the cluster head node reuses the 5G communication frequency band, and co-channel interference may occur between the links. Other sensors in the cluster share the entire bandwidth with the link between the UAV and the cluster head node. Since the simplified Line-of-Sight (LoS) channel model ignores the effects of multi-path fading and shadowing, this channel model is not accurate in areas with many obstacles. Therefore, considering the probabilistic LoS channel model, the probability that the link between UAV u and cluster head node $c_{i}$ conforms to the LoS link can be expressed as follows:(2)$$P_{iu}^{\mathrm{LoS}}=\frac{1}{1+a\exp\left[-b\left(\lambda -a\right)\right]}$$Here, both parameters a and b are related to the environment, and λ is the elevation angle between the cluster head node and the UAV, which can be expressed as $\lambda =\frac{180}{\pi }\arctan\left(\frac{H}{d_{iu}}\right)$. Then the average path loss formula between UAV u and cluster head node $c_{i}$ can be expressed as:(3)$$L_{iu}^{\mathrm{avg}}=P_{iu}^{\mathrm{LoS}}L_{\mathrm{LoS}}+\left(1-P_{iu}^{\mathrm{LoS}}\right)L_{\mathrm{NLoS}}$$
where $L_{\mathrm{LoS}}$ and $L_{\mathrm{NLoS}}$ denote the average path loss for LoS and Non-Line-of-Sight (NLoS) links, respectively; their definitions are as follows:(4)$$L_{\mathrm{LoS}}=20\log_{10}\left(\frac{4\pi f_{c}d_{iu}}{c}\right)+T_{\mathrm{LoS}}$$
(5)$$L_{\mathrm{NLoS}}=20\log_{10}\left(\frac{4\pi f_{c}d_{iu}}{c}\right)+T_{\mathrm{NLoS}}$$where $f_{c}$ is the carrier frequency, c is the speed of light, and $T_{\mathrm{LoS}},T_{\mathrm{NLoS}}$ is the average path extra loss for LoS and NLoS links, respectively. Thus, it can be concluded that the received power of UAV u is $P_{c}-L_{iu}^{\mathrm{avg}}$. In order to ensure that UAV can receive data normally, the received power at the receiver needs to be greater than the threshold $P_{th}$. So the channel gain from cluster head node $c_{i}$ to UAV u can be expressed as:(6)$$g_{iu}(t)=L_{iu}^{\mathrm{avg}}d_{iu}^{-\alpha }(t)$$Here, $d_{iu}(t)$ represents the distance between cluster head node $c_{i}$ and UAV u, which can be defined as:(7)$$d_{iu}(t)=\left\|q_{u}(t)-w_{i}\right\|$$

The achievable rate of data collection can be derived from the received power:(8)$$R_{iu}=B\log_{2}\left(1+\frac{P_{c}g_{iu}\left(t\right)}{P_{Iter}\left(t\right)+\sigma^{2}}\right)$$where $P_{Iter}(t)$ is the interference of other cluster head node $c_{i^{\prime }}$ to the communication link of UAV u, and can be defined as:(9)$$P_{Iter}\left(t\right)=\sum_{i^{\prime }=1,i^{\prime }\neq i}^{K}I_{i^{\prime }u}\left(t\right)P_{c}g_{i^{\prime }u}\left(t\right)$$
where $I_{i^{\prime }u}(t)$ is an indicator function of whether other cluster head nodes interfere with the link between UAV u and cluster head node $c_{i}$, and can be defined as:(10)$$I_{i^{\prime }u}(t)=\left\{\begin{array}{l} 1,\left(P_{c}-L_{i^{\prime }u}^{\mathrm{avg}}\right)\geq P_{th}\\ 0,\mathrm{otherwise} \end{array}\right.$$

In the interest of comprehensive analysis, this paper predominantly centers its focus on a single-antenna scenario. Nevertheless, it is important to emphasize that the modeling and algorithms introduced within this paper possess versatile applicability, readily extending to a multiple-antenna scenario, albeit with a commensurate rise in complexity.

For instance, in the context of a multi-antenna system, the representation of transmission power undergoes a fundamental transformation. Instead of a single variable denoting power P, it evolves into a vector P, with each element P[i] corresponding to a distinct antenna i. This expansion of dimensionality necessitates a shift in the way we handle power allocation and resource management. Similarly, the characterization of the communication channel shifts from a simple scalar value g to a matrix g, which contains channel gain information from a node to multiple antennas of a UAV. This transition is a direct response to the intricate interplay among multiple antennas and their respective channel conditions. It underscores the necessity for advanced techniques such as Multiple Input Multiple Output (MIMO) to harness spatial diversity, thereby enhancing overall system performance. Moreover, the complexities related to interference management, beamforming, and signal processing become significantly more intricate when transitioning from a single-antenna to a multi-antenna environment.

In terms of the proposed solutions in the following sections, the sensor node clustering algorithm based on data volume balance is generally unaffected by whether it is applied in a single-antenna or multi-antenna scenario. However, for the multi-UAV data collection trajectory planning algorithm, when operating in a multi-antenna scenario, it necessitates consideration of additional factors. These mainly include the following aspects: Firstly, owing to the multi-antenna system’s capacity to facilitate concurrent communication with multiple targets, a reevaluation of data collection task prioritization and allocation may be imperative. Secondly, the presence of multiple antennas introduces added intricacy to the communication channel’s state. UAVs must factor in variables such as target locations, channel quality, and interference, among others. Thirdly, multi-antenna systems typically incorporate directional antennas, affording UAVs the flexibility to adjust antenna orientation to augment signal transmission. Consequently, trajectory planning must grapple with the optimization of parameters, including the directional angles of multiple antennas, to maximize the utility of this directional capability. Lastly, multi-antenna systems often demand increased energy resources, necessitating trajectory planning to address effective energy management strategies aimed at extending UAV flight durations.

In summation, transitioning from a single-antenna to a multi-antenna system introduces heightened intricacies and challenges into the realm of data collection trajectory planning. These complexities underscore the imperative for further research in this domain to harness the full potential of multi-antenna technology effectively.

Further more, to clarify the analysis and assist in understanding, we list the acronyms and key symbols of this paper and their full names or descriptions in Table 1.

## 3. Problem Formulation

Path discretization can be used to solve the UAV trajectory planning problem [30]. Let $M_{u}$ be the set of clusters served by UAV u and $A_{u}=\left\{D_{u}^{i},\forall i\in M_{u}\right\}$ denotes the total amount of data that needs to be collected by sensors in each cluster of $M_{u}$. $Q_{u}=\left\{q_{u}^{i}=\left[x_{u}^{i},y_{u}^{i},H\right]|\forall i\in M_{u}\right\}$ is the set of hover locations when UAV u collects data, and $W_{u}=\left\{w_{u}^{i}=\left[x_{u}^{i},y_{u}^{i}\right]|\forall i\in M_{u}\right\}$ is the location of the cluster head node served by UAV u. The sets $A_{u}$, $Q_{u}$ and $W_{u}$ are all ordered sets, and the order corresponds one-to-one. The time spent by UAV u consists of two parts: the flight time of UAV traversing each cluster, and the time to collect data after arriving at the hovering location. Assuming that the average flight speed of UAV u is $v_{u}$, then the flight time of UAV u from the i-th hovering position to the (i+1)-th hovering position can be expressed as follows:(11)$$\left\|q_{u}^{i+1}-q_{u}^{i}\right\|/v_{u}$$

The time spent by UAV u collecting data in the i-th cluster can be formulated as follows:(12)$$D_{u}^{i}/R_{iu}$$

Then the total time taken by UAV u can be represented by T(u):(13)$$T\left(u\right)=\sum_{i\in M_{u}}\left(\left\|q_{u}^{i+1}-q_{u}^{i}\right\|/v_{u}+D_{u}^{i}/R_{iu}\right)$$

According to the above analysis, it is necessary to optimize the clustering of sensors and the location of cluster head nodes, the set of service clusters for each UAV, the hovering position of each UAV, and the order of visiting these position nodes to minimize the total working time of the UAV and balance the service time of each UAV. Therefore, the problem can be formulated as follows:$$\begin{array}{l} P1:\;\underset{S_{i},M_{u},Q_{u},W_{u}}{\min}\underset{u}{\max}\;T(u)\\ s.t.C1:\;\gamma_{c_{i}}>\gamma_{th}^{C},\forall i=1,\ldots ,K\\ \;C2:\;\left(P_{c}-L_{iu}^{\mathrm{avg}}\right)>P_{th},\forall i\in M_{u},u=1,\ldots ,U\\ \;C3:\;\left|\cup_{u=1}^{U}M_{u}\right|=K,\cap_{u=1}^{U}M_{u}=\emptyset ,\forall u=1,\ldots ,U\\ \;C4:\;\left|\cup_{i=1}^{K}S_{i}\right|=N,\cap_{i=1}^{K}S_{i}=\emptyset ,\forall i=1,\ldots ,K\\ \;C5:\;q_{u}^{0}=q_{u}^{\left|M_{u}\right|+1},\forall u=1,\ldots ,U\\ \;C6:\;\left[\max T(u)-\min T(u^{\prime })\right]\leq \varepsilon \end{array}$$

Constraint C1 means that the received SNR of the cluster head node must be greater than the threshold, and constraint C2 means that the received power of the UAV in the hovering position must be greater than the reception threshold. Constraints C3 and C4 indicate that the elements within the cluster are disjoint, and the set of clusters served by each UAV is also disjoint. Constraint C5 indicates that all UAVs must return to the starting position for recharging after collecting data. Constraint C6 states that the difference between the longest and the shortest working time of the UAV after a period needs to be less than the threshold ε. Note that the problem P1 is a non-convex combinatorial optimization problem with nonlinear constraints, which is difficult to solve using traditional mathematical methods, so it is necessary to design an effective new algorithm.

## 4. Problem Solution

In this section, we propose a two-stage approach to solve the problem. Firstly, the traditional k-means clustering method is improved according to the amount and type of sensor data in the cluster to balance the task data in each cluster. Secondly, after determining the clustering situation, the data collection trajectory optimization problem for each UAV is treated as an independent TSP, solved using a grouped improved genetic algorithm, and a 2-opt optimization operator is introduced to improve the convergence efficiency of the algorithm.

### 4.1. Task-Balanced Sensor Clustering Algorithm

Usual clustering algorithms have no restrictions on the number and type of elements in each cluster, and the number of cluster members in the clustering scheme is usually random [31,32,33]. As a result, clusters can vary widely in the number and type of elements. In our scenario, this will result in an uneven amount of data within each cluster, leading to an imbalanced working time for UAVs. Moreover, the data types transmitted by the sensors include three types: simple data stream, audio and video, which are denoted by $D^{s},D^{a},D^{v}$, respectively. The clustering method is divided into two steps: first, through the original k-means clustering method, a partition method of K clusters that satisfies the conditions is found. Second, the number and types of sensors in the cluster were adjusted according to the requirements, and the elements and clusters in the cluster were removed after adjusting a cluster until the last cluster was adjusted. The detailed process of the two steps is shown in Algorithms 1 and 2.

Step 1: Divide the sensors into K clusters using the k-means algorithm

**Algorithm 1:** k-Means clustering algorithm.
Initializes the distance threshold between the cluster head node and other nodes in the cluster, which is defined as $d_{s}^{th}={\left(\frac{P_{s}}{\gamma_{th}^{C}\sigma^{2}}\right)}^{\frac{1}{\alpha }}$.
Randomly select K samples as the cluster head nodes.
Calculate the distance of each sample and group it to the nearest cluster head node, resulting in $S_{i}$ and W.
Calculate the farthest distance $d_{\max}$ between the location of the node in the cluster and the cluster head node.
While $d_{\max}>d_{s}^{th}$, do:
Repeat 2, 3, 4.
End while.
Generate a set of $S_{i}$ and W that meet the requirements.
End.



Step 2: Adjust the number of sensors and the type of sensors in the cluster.

**Algorithm 2:** Cluster partitioning adjustment algorithm.
Input $S_{i}$ and W obtained by Algorithm 1, the number of sensors in each cluster $\left\{m_{1},\ldots ,m_{K}\right\}$, the type of each sensor, and the number of three types of sensors in each cluster $D_{i}^{s},D_{i}^{a},D_{i}^{v}$.
Calculate the center point O of all cluster head node positions; calculate the expected amount of task data $E\left(D_{i}\right)$ in each cluster and the expected number of three types of sensors $E\left(D_{i}^{s}\right),E\left(D_{i}^{a}\right),E\left(D_{i}^{v}\right)$ in each cluster according to the total number of the three kinds of sensors.
Calculate the distance between all cluster head nodes and the center position O, and start adjusting with the outermost cluster.
Compare the actual task data volume within the cluster with $E\left(D_{i}\right)$. If it is less than $E\left(D_{i}\right)$, supplement sensor nodes from other clusters to this cluster; if it is greater than $E\left(D_{i}\right)$, remove some sensor nodes from the cluster.
The order of supplementing sensor nodes is based on proximity in terms of location and follows the order of $D^{s},D^{a},D^{v}$ in terms of types. Once the number of sensor nodes of one type exceeds the expected quantity, the next type is supplemented until the cluster’s data load reaches $E\left(D_{i}\right)$.
The order for removing sensor nodes is based on location, starting from farthest to nearest, and in terms of types, it follows the order of $D^{s},D^{a},D^{v}$. Once the number of sensor nodes of one type falls below the expected quantity, the next type is removed until the data load reaches below $E\left(D_{i}\right)$.
After updating all elements of the cluster, set K=K−1.
Until K=1.
Update the new cluster head node position for each cluster after the adjustment is completed.
Output the adjusted $S_{i}$ and W.



### 4.2. Stepwise Combinatorial Optimization to Solve the MTSP

The problem P1 needs to optimize the clustering of sensors, the location of cluster head nodes, the set of clusters served by each UAV, the position of each UAV hovering, and the order of visiting these location nodes. In Section 4.1, the sensor clustering algorithm with balanced task load has solved the sensor clustering problem well and found the location of the UAV collection. The remaining problem is to solve the division of the UAV service cluster and the order of serving these clusters, which can be abstracted as a MTSP. Each UAV is a traveling salesman, and each cluster head node location is a city.

MTSP is a classical combinatorial optimization problem, and there are many mature solutions [34,35,36]. However, when the number of cities is large, the complexity is high. Specifically, when the number of cities is more than 20, the convergence efficiency is very low; when the number of cities is more than 30, some of them may fail to converge. Therefore, to be applicable to our scenario, a distribution optimization approach is adopted. Firstly, the original MTSP is decomposed into independent subproblems by k-means according to the number of UAVs. Secondly, for each independent Traveling Salesman Problem (TSP), an improved genetic algorithm based on 2-opt optimization is used to solve it.

In our problem scenario, all cluster head nodes are firstly grouped twice, and the ordinary k-means algorithm is used to decompose the MTSP into independent TSPs according to the number of UAVs. This part can be directly referred to as the ordinary k-means of Algorithm 1. Because the convergence speed of the ordinary genetic algorithm will be slow in the larger scale TSP, the improved genetic algorithm with a 2-opt operator is used to solve the TSP. The process is shown as follows (Figure 2).

When the scale of the problem is large, it is prone to an exponential explosion. For this, we mainly refer to the method in [37] and introduce the 2-opt optimization operator. Before the genetic operation, the inferior individuals are locally optimized separately, and some obvious incorrect arrangements are eliminated to avoid invalid operations on these individuals. This method can significantly improve the efficiency of genetic algorithms (Figure 3).

The inverse of the objective function of problem P1 is selected as the fitness function. In order to ensure the dominant characteristics of individuals and avoid premature at the same time, the selection operator is composed of two selection methods in a certain proportion. Among them, in the first part, according to the roulette wheel way, the probability of an individual being selected is positively correlated with the value of the fitness function, and the higher the individual fitness, the higher the probability of being selected. If there are Z individuals and the fitness function value of individual i is $f_{i}$, then the probability that individual i is selected can be expressed as:(14)$$P_{i}=\frac{f_{i}}{\sum_{z=1}^{Z}f_{z}}$$

The selection operator in this part accounts for 90%. In the second part, the doping operator is used to randomly generate new individuals and directly join the next generation population to maintain the diversity of the population, which accounts for 10%. The crossover operator is not the usual way of choosing two-point crossover, because this way requires conflict detection, which increases the computational burden and may perturb the originally correct gene segments in the uncrossed part. The crossover operator draws two chromosomes as parents, where one parent 1 randomly selects the start and stop positions of the segments, and the genes of the crossover segments of the offspring are the same as those of the parent 1. In parent generation 2, the crossed gene fragments are removed, and the remaining genes are put into the offspring in the original order, which produces a new offspring. The mutation operator randomly extracts the start and end positions of the chromosome and rearranges the mutated segments.

### 4.3. Analysis of Complexity

Problem P1 forms a MTSP, which means that a certain number of target points are assigned to multiple traveling agents, and each traveling agent starts from the starting point, passes through the assigned target points in turn, and finally returns to the starting point. MTSP is a NP-hard combinatorial optimization problem. In our scenario, with n cluster head nodes and k UAVs, the time complexity obtained by using the exhaustive method is $O\left(\frac{n!}{k\times (n-k)!}\right)$; when n and k grow, the exponential explosion occurs, and the calculation cannot be completed in practical time even with the fastest computer. However, when a grouping genetic algorithm is used to solve the traveling salesman problem, the approximate optimal solution can be obtained in an acceptable time. Assuming that the size of the single TSP problem obtained by grouping is $\frac{n}{k}$, the population size is m, and the number of iterations is L, then the time complexity of grouping genetic algorithm to solve the MTSP is $O\left(\frac{n}{k}\times m\times L\right)\times O\left(k\right)=O\left(nmL\right)$, which is mainly related to the population size and the number of iterations. Moreover, 2-opt is used as the optimization operator. The experimental results in the next section also show that the number of iterations L is effectively reduced after optimization, which further reduces the time complexity of the algorithm.

## 5. Simulation and Analysis

Next, we will prove the feasibility of the algorithm through simulation. Firstly, some parameter conditions used in the simulation are given. Referring to the simulation settings in some of the relevant prior studies [37,38,39] and taking into account the specific scenario of our research, our simulation parameters are presented. Here we suppose the number of UAVs in the area U=3, and the number of sensor nodes in the whole network N=300. These sensors are randomly distributed in the 2-D space of 2 km×2 km. The flying height of all UAVs is assumed to be fixed at 100 m in the air. For the channel parameters, the power of AWGN additive white noise $\sigma^{2}$ is set to −120 dBm, the parameters in the probabilistic LoS link model are set to a=9.53 and b=0.41, the receiving threshold $\gamma_{th}=0\;\mathrm{dB}$ at the UAV receiver, and the maximum transmit power of the cluster head node is $P_{c}=10\;\mathrm{dBm}$. And the maximum speed of the UAV $v_{\max}=30\;m/s$. The detailed information on all simulation parameters is shown in Table 2.

Figure 4 shows the initial distribution of sensors, with a total of 300 sensors. The types of these sensor devices are classified into three types: data streams, audio, and video. The data volume of these three types of sensors is 0–10 MB, 10–20 MB, and more than 100 MB, respectively. In Figure 4, black represents the data stream sensor, blue represents the audio sensor, and red represents the video sensor. Figure 5 shows the distribution of cluster head nodes obtained by sensor clustering after running Algorithms 1 and 2. The sensors are divided into 30 clusters, and the amount of data to be collected in each cluster is different.

Figure 6 and Figure 7 show the amount of data within each cluster using the traditional k-means algorithm and our proposed clustering algorithm, respectively. From these two bar charts, it can be seen that the data amount in the cluster results using the traditional k-means algorithm is significantly unbalanced. For example, in the 8-th and 9-th clusters, the amount of data that needs to be collected is under 200 MB, while in the 27-th cluster, the amount of data is approaching 1 GB. In the results obtained by using the clustering algorithm with balanced task volume, the gap is obviously reduced, which can effectively balance the working time of each UAV data collection.

Figure 8 and Figure 9 show the paths obtained by solving the MTSP based on the original GA and grouping the improved GA, respectively. The varying trajectories, each distinguished by its distinct color, depict the flight paths of individual UAVs. The total distance of the improved algorithm after grouping is obviously better than that of the original genetic algorithm because the genetic operator is improved by using the 2-opt method, and the poor individuals are eliminated from the evolution. It can be seen that the best and longest path length based on the original genetic algorithm is 8374.9031 m, while the result of the improved genetic algorithm by grouping is only 6134.0561 m.

In addition, due to the group solution of the original MTSP, the convergence rate of our proposed algorithm is also faster. Figure 10 shows the convergence of the two algorithms. When using the original genetic algorithm, it converges after 80 iterations, while when the grouped improved genetic algorithm is used, it can be seen from the figure that the algorithm converges after 40 iterations. Compared with the original genetic algorithm, the convergence speed is greatly improved.

Figure 11 shows the comparison of UAV running times. It can be seen that through our method, the longest UAV running time, that is, the total time of task completion, is obviously optimized. Moreover, due to the cluster design and balanced task volume, the running time of each UAV is not significantly different.

Figure 12 shows the task completion time under different cluster numbers. At the scale of 300 sensors, it is better divided into 30 clusters. When the number of clusters is small, the number of sensors in each cluster increases, and the bandwidth allocated to the link between the cluster head node and the UAV decreases, resulting in a longer data collection time. When the number of clusters increases, the path of the UAV becomes longer, and the flight time becomes the main factor leading to the increase in task completion time.

In order to better validate the effectiveness of the method, a control experiment was conducted under different numbers of sensors, randomly generating 100–600 sensors. The original genetic algorithm and the grouped, improved genetic algorithm were used to compare the completion time of the task. The results in Figure 13 show that the original genetic algorithm has been able to obtain a better solution in the case of a small sensor scale, and the improved method has a limited reduction in task completion time in this case. When the number of sensors is large and the scale of the problem is large, the optimization algorithm can significantly improve the task completion time.

## 6. Conclusions

In this paper, we present a novel data collection mechanism aimed at minimizing task completion time for multi-UAV cooperation in the IoT scenario of large-scale terminals, where most sensors have low transmission power, no storage capacity and energy consumption constraints. To address these challenges, a two-stage processing method is proposed. Firstly, all sensors are organized into clusters, with cluster head nodes responsible for data collection within each cluster. Subsequently, employing multiple UAVs, we devise their trajectories to efficiently collect data from the cluster head nodes. In order to balance the running time of all UAVs and the workload, a task load balancing clustering algorithm is proposed. Additionally, we apply a path discretization technique to address UAV trajectory planning and employ a grouping improved genetic algorithm to design the path for each UAV. Our simulation results confirm the efficacy and superiority of the proposed algorithm. While this paper primarily addresses the challenges of data collection within WSNs using multiple UAVs, there is a promising avenue for future research. This entails exploring collaborative data processing and analysis mechanisms among UAVs to further enhance data processing efficiency after they finish data collection. Additionally, given the inherent energy constraints in WSN devices, the development of energy-efficient data collection strategies emerges as a critical imperative. Furthermore, bolstering the levels of security and privacy in WSN data collection remains paramount, suggesting future research could delve into the realm of secure communication protocols and advanced data encryption techniques.

## Figures and Tables

**Figure 1 sensors-23-08601-f001:**
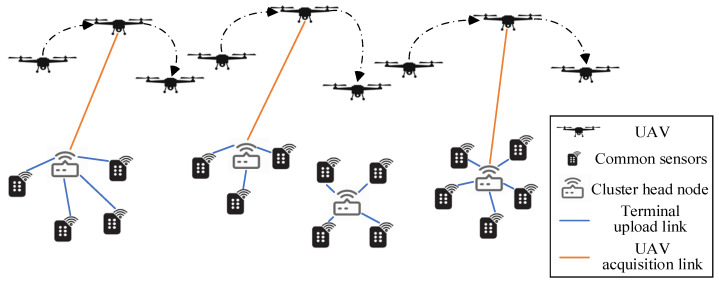
Multi-UAV data collection scenario for large-scale terminal access.

**Figure 2 sensors-23-08601-f002:**
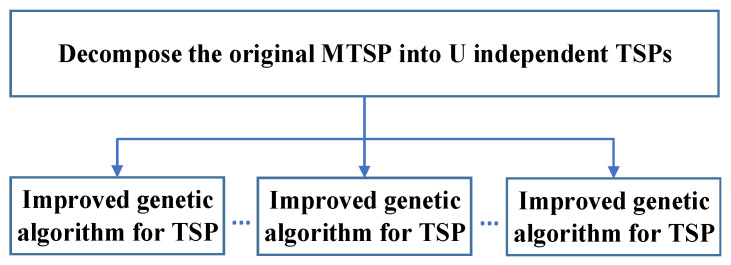
Distribution combination process.

**Figure 3 sensors-23-08601-f003:**
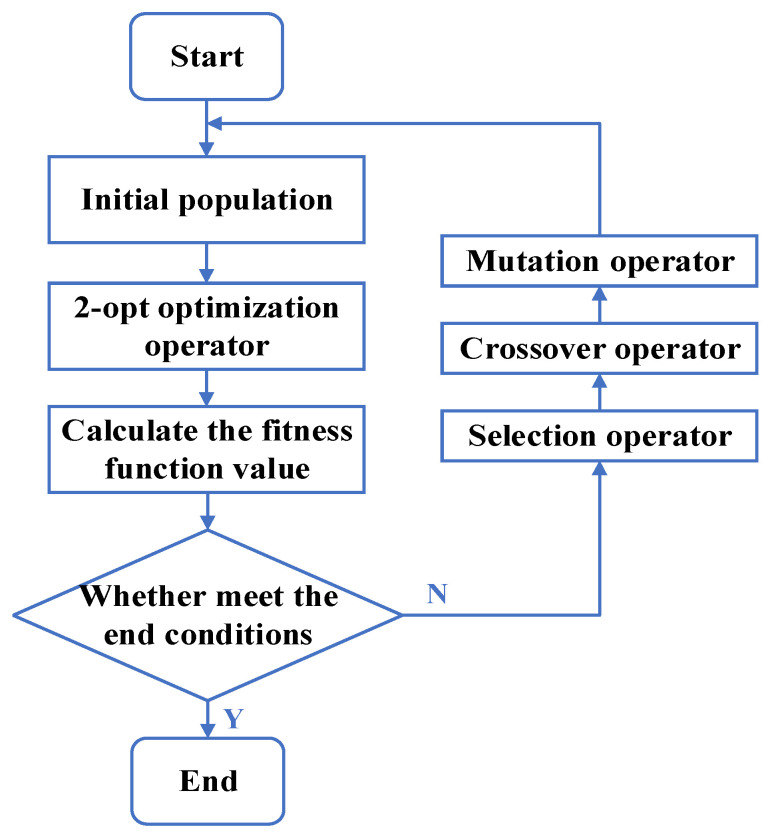
Genetic algorithm based on 2-opt optimization.

**Figure 4 sensors-23-08601-f004:**
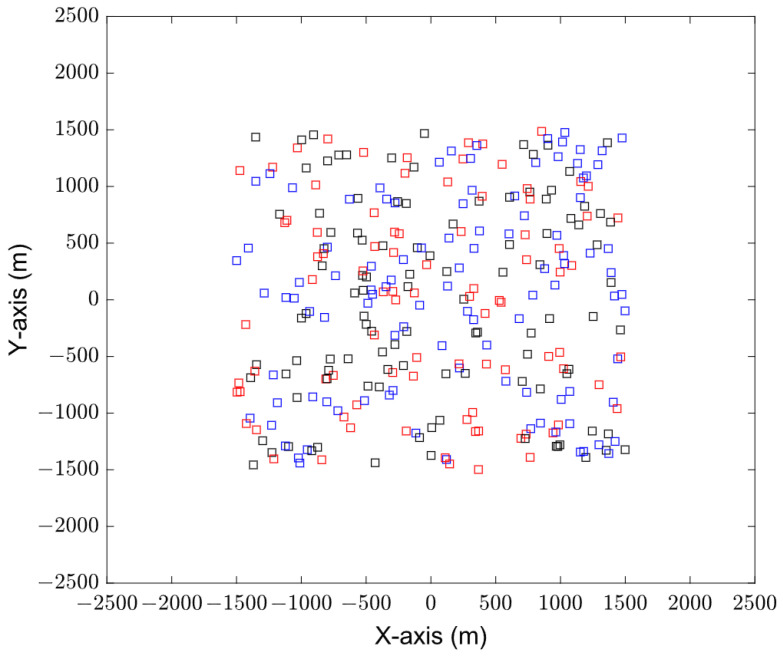
Sensor initial distribution.

**Figure 5 sensors-23-08601-f005:**
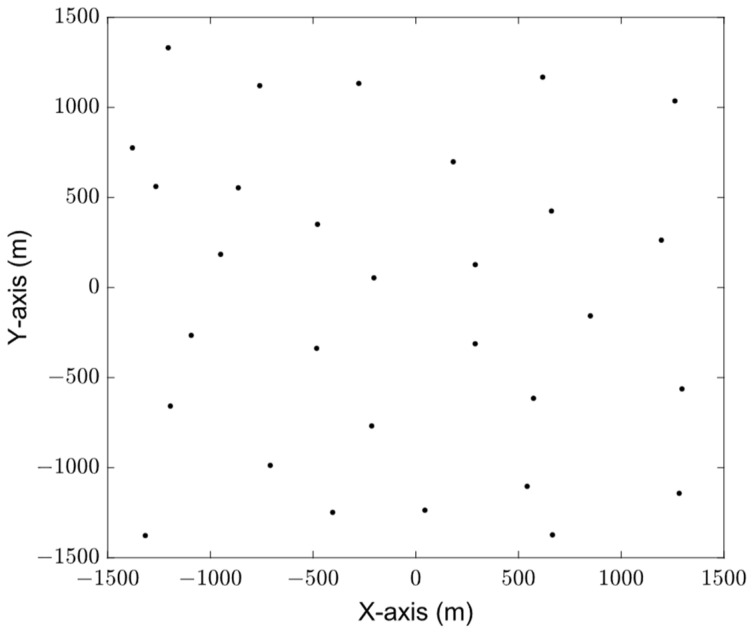
Location of cluster head nodes.

**Figure 6 sensors-23-08601-f006:**
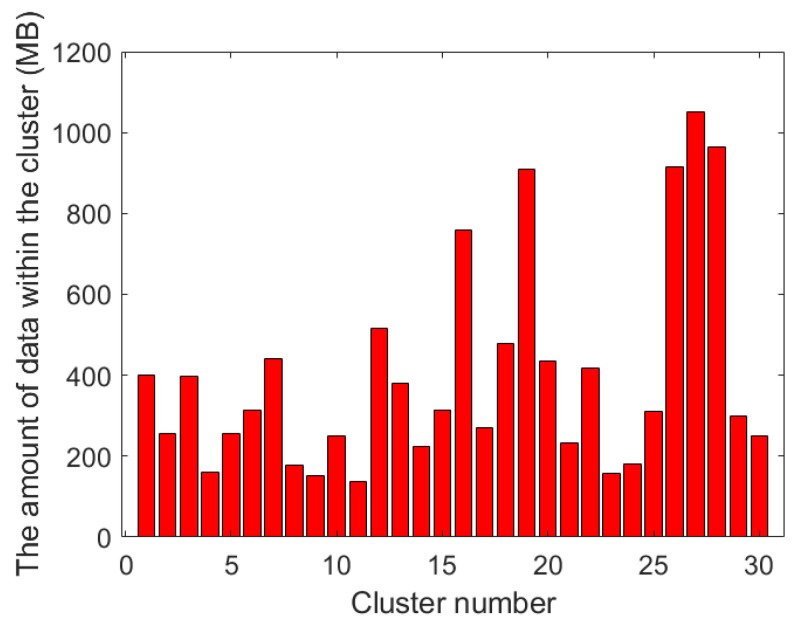
Data volume of each cluster of the original k-means algorithm.

**Figure 7 sensors-23-08601-f007:**
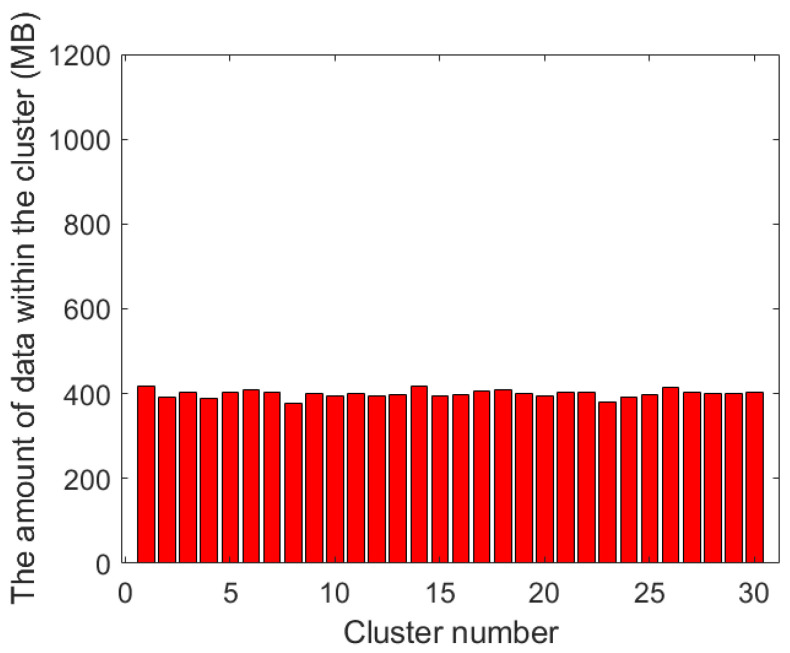
Data volume of each cluster in the clustering algorithm with balanced task volume.

**Figure 8 sensors-23-08601-f008:**
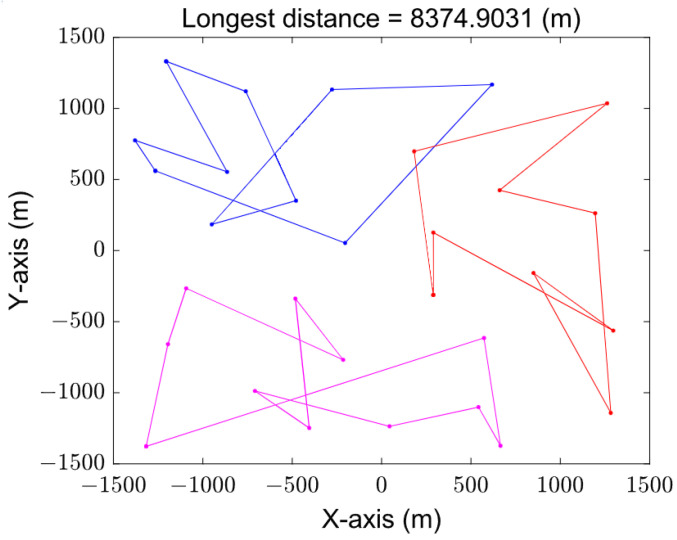
Trajectory obtained by original GA.

**Figure 9 sensors-23-08601-f009:**
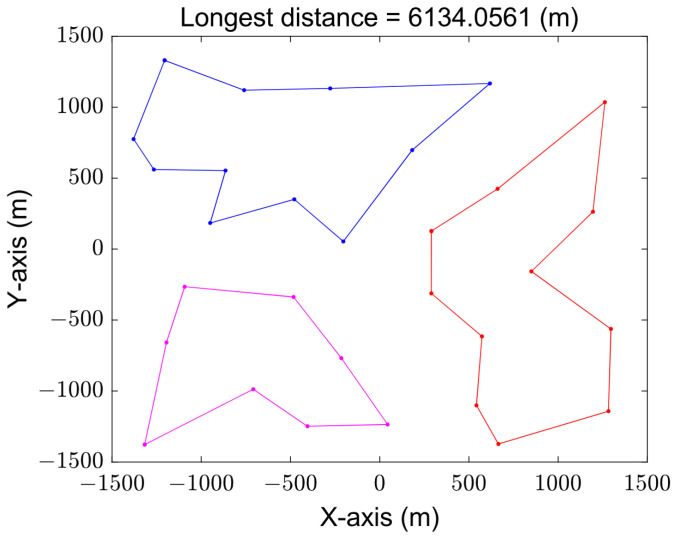
Trajectories obtained by grouping improved GA.

**Figure 10 sensors-23-08601-f010:**
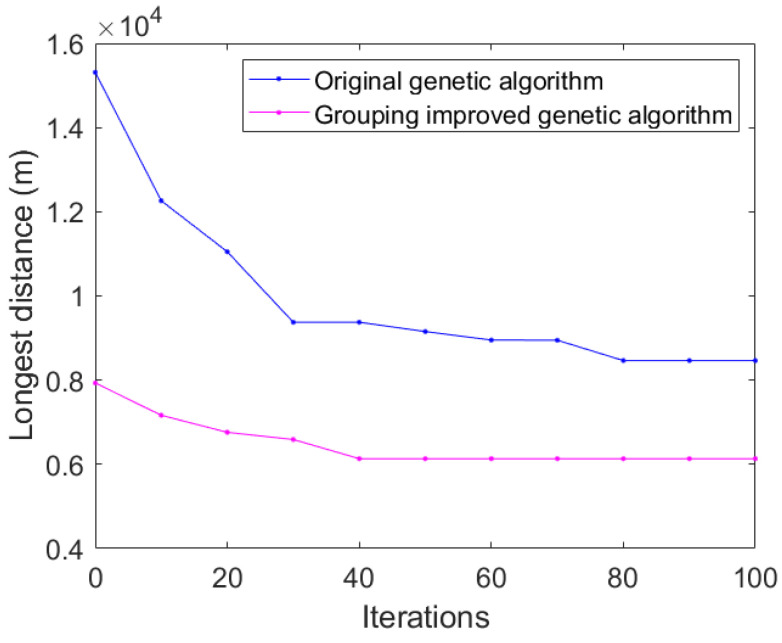
Comparison of algorithm convergence.

**Figure 11 sensors-23-08601-f011:**
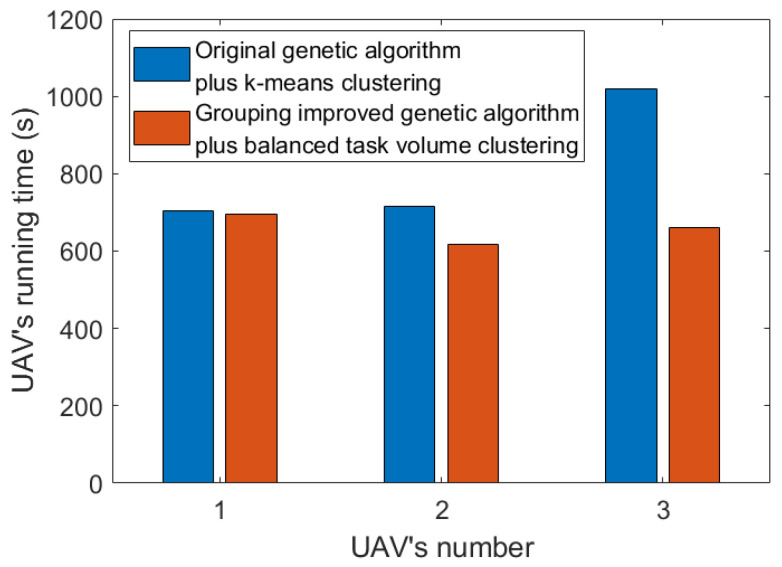
Comparison of UAV running time.

**Figure 12 sensors-23-08601-f012:**
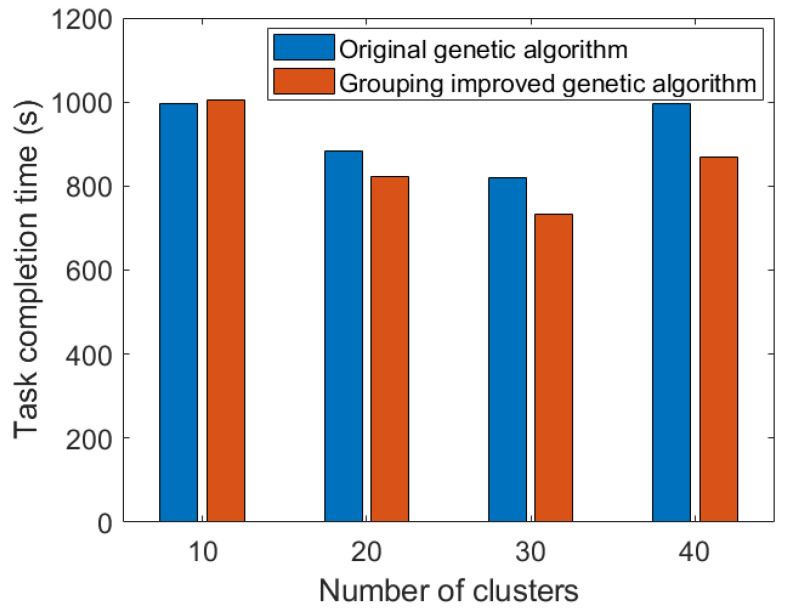
Comparison of completion time under different numbers of clusters.

**Figure 13 sensors-23-08601-f013:**
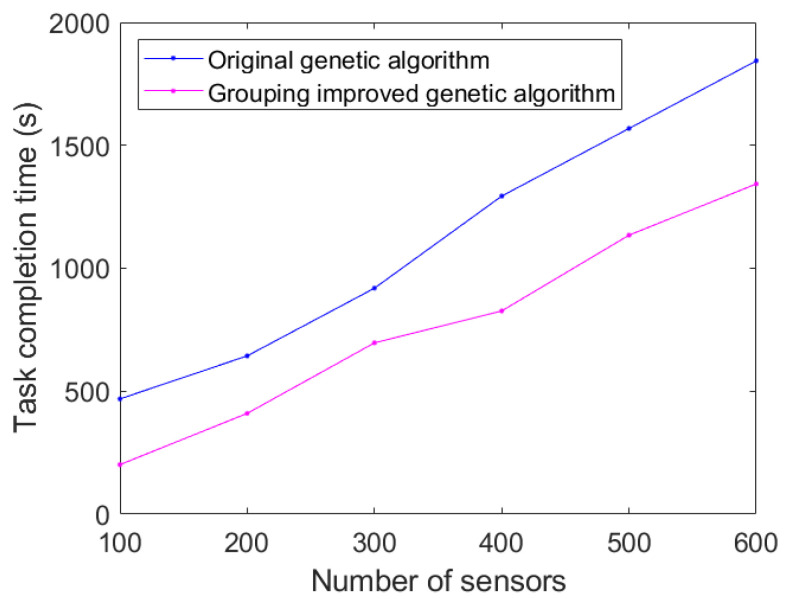
Comparison of task completion time under different sensor sizes.

**Table 1 sensors-23-08601-t001:** List of acronyms and key symbols.

Acronyms/Symbol	Full Name/Description
AWGN	Additive White Gaussian Noise
GA	Genetic Algorithm
IoT	Internet of Things
LoS	Line-of-Sight
MTSP	Multiple Traveling Salesman Problem
NLoS	Non-Line-of-Sight
RSS	Received Signal Strength
SNR	Signal-to-Noise Ratio
TSP	Traveling Salesman Problem
UAV	Unmanned Aerial Vehicle
UGV	Unmanned Ground Vehicle
WSN	Wireless Sensor Network
*N*	Number of sensor nodes
*K*	Number of cluster head nodes
*U*	Number of UAVs
$P_{c}$	The transmit power of the cluster head node
$P_{s}$	The transmit power of other sensor nodes
$P_{th}$	The threshold of the received power
$c_{i}$	A cluster head node
$s_{j}$	A sensor node
u	A UAV
$\gamma_{c_{i}}$	The received SNR of cluster head node $c_{i}$
$\gamma_{th}^{C}$	The threshold of the received SNR
$d_{ji}$	Distance between sensor node $s_{j}$ and cluster head node $c_{i}$
α	The path loss exponent
$\sigma^{2}$	The AWGN noise power
$T_{u}$	Completion time for UAV u to complete data collection tasks
H	The UAV flight altitude
$q_{u}(t)$	The horizontal trajectory position of UAV u
$S_{i}$	The set of all sensor nodes in the i-th cluster
W	The set of the positions of all cluster head nodes
$P_{iu}^{\mathrm{LoS}}$	The probability of the link between cluster head node $c_{i}$ and UAV u conforming a LoS link
λ	The elevation angle between cluster head node $c_{i}$ and UAV u
$L_{iu}^{\mathrm{avg}}$	The average path loss between cluster head node $c_{i}$ and UAV u
$L_{\mathrm{LoS}}$	The average path loss for LoS links
$L_{\mathrm{NLoS}}$	The average path loss for NLoS links
$g_{iu}(t)$	The channel gain between cluster head node $c_{i}$ and UAV u
$d_{iu}(t)$	The distance between cluster head node $c_{i}$ and UAV u
$R_{iu}$	The achievable rate of data collection between cluster head node $c_{i}$ and UAV u
$P_{Iter}(t)$	The interference of other cluster head nodes to the communication link of UAV u
$I_{i^{\prime }u}(t)$	An indicator function of whether other cluster head nodes interfere with the link between cluster head node $c_{i}$ and UAV u
$M_{u}$	The set of clusters served by UAV u
$A_{u}$	The total amount of data that needs to be collected by sensors in each cluster of $M_{u}$
$Q_{u}$	The set of hover locations when UAV u collects data
$W_{u}$	The location of the cluster head nodes served by UAV u
$v_{u}$	The average flight speed of UAV u
T(u)	The total time taken by UAV u
$D_{i}$	The amount of task data in the i-th cluster
$D^{s},D^{a},D^{v}$	The three data types transmitted by the sensor nodes: simple data stream, audio and video
$d_{s}^{th}$	The distance threshold between the cluster head node and other nodes in the cluster
$E\left(D_{i}\right)$	The expected amount of task data in the i-th cluster
$E\left(D_{i}^{s}\right),E\left(D_{i}^{a}\right),E\left(D_{i}^{v}\right)$	The expected number of three data types in the i-th cluster
Z	Size of population
$f_{i}$	The fitness function value of individual i
$P_{i}$	The probability that individual i is selected

**Table 2 sensors-23-08601-t002:** Simulation Parameter Settings.

Simulation Parameters	Parameter Settings
UAV flight altitude H	100 m
AWGN noise power $\sigma^{2}$	−120 dBm
LoS link probability model parameter a,b	9.53, 0.41
UAV reception threshold $\gamma_{th}$	0 dB
The maximum transmit power of the cluster head node $P_{c}$	10 dBm
The maximum speed of the UAV $v_{\max}$	30 m/s
Total system bandwidth	240 MHZ
Size of population	80
Maximum number of iterations	100
Probability of crossover	0.65
Probability of mutation	0.25

## Data Availability

No additional data are available.

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
