# Peer review of "Multi-UAV Data Collection and Path Planning Method for Large-Scale Terminal Access"

_sensors, 2023, doi:10.3390/s23208601_

Round 1
Reviewer 1 Report
Please see the attached comments.

Minor editing of English language is needed.
Reviewer 2 Report
This paper focuses on optimizing the complex interplay between task completion time and task volume equilibrium. To this end, a novel strategy is devised that integrates sensor area partitioning and flight trajectory planning for multiple UAVs, forming an optimization framework geared towards minimizing task completion duration. The core idea of this endeavor involves devising an innovative adaptation of the k-means algorithm, thoroughly calibrated to equalize sensor data distribution across clusters, thus fostering balanced data allocation. Then, the UAV flight trajectory paths are discretely modeled and the grouped 2-opt improved genetic algorithm is used to solve the formed multi-traveler problem. Empirical validation through comprehensive simulations clearly underscores the efficacy of the proposed approach. In particular, the method demonstrates a remarkable capacity to rectify the historical issue of diverse task volumes among multiple UAVs, all the while significantly reducing task completion times. Moreover, its convergence rate substantially outperforms that of the conventional genetic algorithm, attesting to its computational efficiency. This paper contributes an innovative and efficient paradigm to improve on the problem of data collection from IoT terminals through the use of multiple UAVs. As a result, it not only augments the efficiency and balance of task distribution but also showcases the potential of the tailored algorithm solutions in realizing the optimal outcomes in complex engineering scenarios.
This reviewer has the following concerns.
1. Overall, the paper is written and structured well. The topic of this paper is also interesting and timely. The authors also proposed an optimization framework and provide some numerical results. It would be better to add a table for acronyms and symbols used in this paper.
2. In the proposed model, the authors consider a single antenna scenario which is rather simple as practical systems used multiple antennas. The authors should at least explain how this work can be extended to a multiple antenna scenario and what changes will appear in the system, mathematical modeling, and proposed solution?
3. The path loss and channels used in this paper follows 3GPP? Please justify it if not.
4. The introduction section is weak and need to be further improved. For instance, the literature work missing several related works. By searching the recent work, I found the following related works: Device-Free Motion & Trajectory Detection via RFID, A Composite Adaptive Fault-Tolerant Attitude Control for a Quadrotor UAV with Multiple Uncertainties, Hierarchical Velocity Optimization for Connected Automated Vehicles With Cellular Vehicle-to-Everything Communication at Continuous Signalized Intersections, A Novel Airspace Planning Algorithm for Cooperative Target Localization, UAV-Assisted Task Offloading in Vehicular Edge Computing Networks, Understanding Private Car Aggregation Effect via Spatio-Temporal Analysis of Trajectory Data, On-Ramp Merging Strategies of Connected and Automated Vehicles Considering Communication Delay, Integrated Sensing and Communications for UAV Communications with Jittering Effect, Flight trajectory prediction enabled by time-frequency wavelet transform. Nature Communications, Multi-UUV Maneuvering Counter-Game for Dynamic Target Scenario Based on Fractional-Order Recurrent Neural Network, Trajectory optimization of an electric vehicle with minimum energy consumption using inverse dynamics model and servo constraints, Many-Objective Deployment Optimization for a Drone-Assisted Camera Network.
5. Study, discuss these works carefully. Then provide comparative analysis and explain how the proposed work is different from the existing literature.
6. Fig. 2, and Fig.3 are not visible. The authors need to improve the quality of theses figures. Also, the font size in these figures is small than the paper text. Same for the Fig 4-14, their font size is almost invisible. Please revise these figures and improve their visibility.
7. In the concluding remarks, mention some potential research directions for the reader of this paper. Add about how to extend this work in future.
8. In addition, carefully proofread the paper for typos and grammar errors and correct those where needed.
Proofrading is required.
Round 2
Reviewer 1 Report
The authors have well addressed all my concerns, no further comments.
Author Response
Thank you very much for your thorough review and valuable feedback. We greatly appreciate your diligence in assessing our work, and we are delighted to hear that we have successfully addressed all of your concerns. Your expert guidance has been immensely helpful for our research, and we sincerely appreciate your time and effort. We look forward to taking this paper to the next level, and once again, thank you for your support and feedback!
Reviewer 2 Report
The author do not address my comments. I have serious doubt on the novelty of this paper. For example, the literature work is not comprehensive, the author did not report many paper related to this work. Some of the recommended works are also ignored. Moreover, this work consider only single model with single antenna secenario, it is not practical. The font size is still very small. I am not recommending this paper for publication as it is still required improvement. Author should addressed all theses comments along my previous comments.
Need significant efforts.
